# Isolation of Thermophilic Bacteria from Extreme Environments in Northern Chile

**DOI:** 10.3390/microorganisms12030473

**Published:** 2024-02-27

**Authors:** Bernardita Valenzuela, Francisco Solís-Cornejo, Rubén Araya, Pedro Zamorano

**Affiliations:** 1Laboratorio de Microorganismos Extremófilos, Instituto Antofagasta, Universidad de Antofagasta, Antofagasta 1240000, Chile; francisco.solis@uamail.cl; 2Departamento de Educación, Facultad de Educación, Universidad de Antofagasta, Antofagasta 1240000, Chile; 3Instituto de Ciencias Naturales Alexander von Humboldt, Facultad de Ciencias del Mar y Recursos Biológicos. Universidad de Antofagasta, Antofagasta 1240000, Chile; ruben.araya@uantof.cl; 4Departamento Biomédico, Facultad de Ciencias de la Salud, Universidad de Antofagasta, Antofagasta 1240000, Chile

**Keywords:** thermophiles, hydrolytic enzymes, extreme environments, Atacama Desert

## Abstract

The northern region of Chile boasts unique geographical features that support the emergence of geothermal effluents, salt lagoons, and coastal creeks. These extreme climate conditions create polyextreme habitats for microorganisms, particularly adapted to survive these harsh environments. These extremophilic microorganisms hold immense potential as a source of hydrolytic enzymes, among other biotechnological applications. In this study, we isolated 15 strains of aerobic thermophilic bacteria (45–70 °C) from sediment samples collected at five different ecological sites, including hot springs, geothermal fields, and lagoons in the Atacama Desert and Andes high planes. Analyses of the 16S rRNA gene sequences of the isolates showed a close genetic similarity (98–100%) with microorganisms of the genera *Parageobacillus*, *Geobacillus*, *Anoxybacillus*, and *Aeribacillus*. Notably, these thermophiles exhibited significant hydrolytic enzyme activity, particularly amylases, lipases, and proteases. These findings underscore the potential of using these thermophilic bacterial strains as an invaluable source of thermozymes with wide-ranging applications in diverse industries, such as detergent formulations, pharmaceutical processing, and food technology. This research highlights the ecological significance of these extreme environments in the Atacama Desert and Andes high plains, which serve as vital ecological niches housing extremophilic bacteria as a genetic source of relevant thermozymes, promising great potential for innovation in the biotechnology industry.

## 1. Introduction

The proliferation rate of microorganisms in their habitat is significantly influenced by a myriad of chemical and physical environmental conditions. Among the key factors shaping natural environments, four play a particularly crucial role in determining microbial proliferation: temperature, pH, water availability, and oxygen levels [1,2]. “Moderate” environments are essential for sustaining life, encompassing conditions with pH levels close to neutral, temperatures ranging between 20 and 40 °C, atmospheric pressure at 1, and an adequate supply of water, nutrients, and salts. On the other hand, environments deviating significantly from these ranges, either higher or lower, are defined as “Extreme Environments” [3]. It is important to indicate that the term “extreme” is relative and anthropocentric in nature, not necessarily describing something beyond biological understanding [4,5].

In extreme environments, a diverse array of microorganisms thrives, capable of proliferating under various harsh conditions such as extreme temperatures, salinity, pH, and pressure, earning them the name “extremophiles” [6]. The term “extremophile” was first coined by MacElroy in 1974 to describe microorganisms inhabiting such extreme environments [7]. Researchers have also shown great interest in studying the enzymes of these extremophilic microorganisms, primarily due to their significant economic potential in various industrial processes. These enzymes find applications in agriculture, food production, breweries, detergents, textiles, and even stationery [8]. Moreover, there are biotechnological applications in genetic identification and diagnostics [9]. The advancement of this field has been made possible by the isolation of numerous microorganisms from extreme environments and the subsequent extraction and characterization of their extremozymes [10,11,12,13]. These studies have paved the way for tapping into the potential of extremophiles and their enzymes, opening new possibilities for various industries and biotechnological research. Among all the organisms considered extreme, thermophiles have received the most extensive research attention due to the thermoresistance of their internal components [12,13,14,15,16]. By definition, these microorganisms thrive at temperatures above 45 °C, and those that can proliferate above 80 °C are referred to as hyperthermophiles [17]. Remarkably, some hyperthermophiles can even survive temperatures exceeding 100 °C, exemplified by the Archaea *Pyrococcus furiosus* [18] and *Methanopyrus kandleri* [19].

The studies conducted on these microorganisms consistently demonstrate their close association with thermal habitats, such as hot springs, volcanoes, sulfate fields, and hydrothermal vents on the seabed, co-existing with thermophilic microorganisms of both Bacteria and Archaea domains [17].

To thrive in high-temperature environments, thermophilic organisms undergo cellular and molecular adaptations [16,18,19,20]. Consequently, the majority of enzymes produced by thermophilic and hyperthermophilic microorganisms are thermostable, meaning they resist irreversible denaturation and consequently enzyme inactivation at high temperatures. Moreover, these enzymes are thermophilic, exhibiting optimal activity temperatures between 55 and 110 °C [16]. This characteristic makes them of great interest for commercial innovations in biotechnology [21,22]. They find utility in generating intermediate compounds for the food industry, as well as in the detergent industry [23], and they play a crucial role in molecular biology through DNA-modifying molecules [24]. The diverse applications of these thermophilic enzymes underline their significance and potential for various practical uses.

The geographical characteristics found in the northern Chilean desert and high planes show a favorable environment, with the formation of geothermal springs, saline lagoons, and wetlands near the coast. These aquatic systems host microorganisms with unique adaptations to endure extreme environmental conditions. Given these extraordinary conditions, the Atacama Desert has become a significant area to explore for extremophilic microorganisms with potential biotechnological importance, which remains relatively unexplored in the region.

The main objective of this study was to axenically isolate and characterize thermophilic microorganisms from this particular aquatic ecosystem in northern Chile. The goal was to determine the phylogenetic and biochemical characteristics of these microorganisms and evaluate the biotechnological potential.

The study targeted several locations in northern Chile for thermophile isolation: (i) the El Tatio geothermal field, the world’s third-largest and the Southern Hemisphere’s largest geothermal field [25]; (ii) the Jurasi Hot Springs in the Putre, Arica and Parinacota regions; (iii) Laguna Tebenquiche, within the Atacama salt flat of the second region, where halophilic microorganisms have been previously isolated [26]; (iv) Laguna Cejar, also in the Atacama salt flat, fed by volcanic groundwater with a salinity of 292 g/L [27]; and (v) Quebrada Carrizo, located at the southern entrance of Antofagasta city, approximately 1.5 km from the coast. The study successfully obtained 15 strains of axenic cultures. Through phylogenetic characterization via 16S rRNA gene sequencing analysis, these strains were classified into four genera: *Anoybacillus*, *Aeribacillus, Parageobacillus* and *Geobacillus*. Additionally, various thermozymes, such as amylases, lipases, and proteases, were identified within these isolated strains.

The diverse carbon sources utilized and the enzymatic activities associated with the thermophilic strains isolated in this study suggest a high potential for application in research, and this will contribute to the valorization of the genetic resource diversity in extreme aquatic systems.

## 2. Materials and Methods

### 2.1. Sample Collection

Sludge samples were carefully collected under aseptic conditions from the specified locations shown in Figure 1. Subsequently, the samples were stored at 4 °C in sterile conical tubes to maintain their integrity during transportation, and later processed in the Laboratory of Extremophile Microorganisms at the University of Antofagasta. Measurements of pH and temperature in situ were carried out with the use of a Portable pH Meter (pH/Temp) (HI 99141, Hanna Instruments, Woonsocket, RI, USA).

### 2.2. Culture Media and Isolation of Thermophilic Bacteria

To isolate thermophilic aerobic bacteria, we employed Natural Tatio Media (NTM), composed of a water sample from the El Tatio thermal spring (filtered through a 0.2 μm environmental water filter) and supplemented with 0.25% (*w*/*v*) of yeast extract and 0.20% (*w*/*v*) of peptone (Oxoid Ltd., Basingstoke, UK) [28].

To initiate the cultures, approximately 1 g of sediment from collected samples was inoculated into sterile 15 mL glass tubes containing a maximum of 5 mL of the culture medium. The initial culture temperature was maintained at 60 °C for static cultures for 4 days, using a SHEL LAB incubator (model EI2-2, Sheldon Manufacturing, Inc. 300 N. 26TH Cornelius, OR 97113, USA); while the shaking of cultures at 60 °C for 4 days was performed with a LAB NET shaker incubator (LabNet 311DS, Marshall Scientific, Co., Huntington, WV, USA) operating at 250 rpm. For agar medium, Petri dishes were prepared by seeding the same medium supplemented with 2% (*w*/*v*) agar. The inoculated plates were then incubated at 60 °C in a humid chamber for 48 h. Isolated colonies were obtained and further subcultured successively to ensure the axenic isolation of the distinct strains.

Bacteria isolated in this medium were then transferred to various culture media, including TT, YPS, Miller, Kunaemneni, Huang, ATM1, and ATM2 (Table 1). The objective was to establish an optimal culture medium with well-known components that can be replicated under laboratory conditions.

### 2.3. Determination of Cultivation Parameters

The growth of isolated bacteria was investigated in different temperature and pH ranges. Inoculum size was determined through the cell counting technique, employing the 4’, 6-diamidino-2-phenylindole (DAPI) stain [29] in a fresh culture. The inoculation of isolated strains was performed using fresh cultures, and the cell count was conducted based on a minimum of 300 cells observed in at least 20 random fields per sample [30]. The strains were cultured in triplicates of 3 mL of liquid ATM1 medium, exposed to temperatures ranging from 37 to 80 °C (37, 40, 45, 50, 55, 60, 65, 70, 75, and 80 °C), and subjected to different pH conditions using various buffers (50 mM Acetic Acid-Acetate for pH 5, 50 mM HEPES for pH 6 and 7, 50 mM Tris-HCl for pH 8, and 50 mM NaHCO_3_-Na_2_CO_3_ for pH 9 and 10). This was done to determine the optimal pH and growth temperature for the strains. Each pH point under evaluation consisted of three replicated culture tubes, including three additional replicates for the control without inoculum.

The optimal growth temperature was determined by inoculating 10,000 cells in 5 mL of liquid medium, which were then placed in capped glass tubes and incubated in thermoregulated baths at the indicated temperatures. Measurements were taken in triplicates for each temperature, and bacterial growth was monitored using a light spectrometer (Evolution 60, Thermo Fisher Scientific, Waltham, MA, USA) at a wavelength of 600 nm.

### 2.4. Determination of Hydrolytic Enzymes Activities

The hydrolytic enzyme activities of the bacterial isolates were determined in ATM1 medium. When activities were assayed in ATM1 agar plates, incubations were carried out in a humid chamber at the optimal temperature for best colony growth. The methodology described in “Methods for General and Molecular Bacteriology” [31] was followed for these characterizations, which are detailed below.

#### 2.4.1. Hydrolysis of Sugars

This was evaluated in ATM1 liquid medium supplemented with 2% (*w*/*v*) of the specific sugar being analyzed (glucose, sucrose) and phenol red as an indicator. After incubation at the optimal temperature for 24 h, a change in the pH indicator from orange to yellow (pH less than 6) was considered a positive result.

#### 2.4.2. Starch Hydrolysis

To determine this, the selected strains were inoculated on plates containing ATM1 media in agar plates with 2% (*w*/*v*) starch, followed by incubation for 4 days. Subsequently, the plates were developed with iodine solution. A translucent halo around the colonies indicated positive amylolytic activity, indicating starch hydrolysis on the plate.

#### 2.4.3. Lipid Hydrolysis

This was determined by inoculating the strains on agar plates with ATM1 media supplemented with 2% *w*/*v* unsalted butter. The strains were spread by streaking and grown at the optimal temperature for each strain under study for 4 days. Afterward, the plate was flooded with a 1% *w*/*v* solution of copper sulfate. Colonies with a light blue precipitate around them were considered positive for lipolytic activity.

#### 2.4.4. Protein Hydrolysis

To assess protein hydrolysis, the selected strains were inoculated on agar plates with ATM1 media supplemented with 5% *v*/*v* skim milk, and incubated at the optimal temperature for 4 days. A translucent halo around the colonies was considered a positive result, indicating proteolytic activity.

#### 2.4.5. Cellulolytic Hydrolysis

This was assessed by inoculating the strains on agar plates with ATM1 media supplemented with 1% *w*/*v* carboxymethylcellulose (CMC). The strains were evenly spread by streaking and incubated at the optimal temperature for each strain under study for 4 days. Subsequently, the plate was inundated with a 0.05% *w*/*v* solution of Congo Red. A translucent halo around the colonies signified positive cellulase activity, indicating starch hydrolysis on the plate.

### 2.5. Sporulation Tests

To determine the presence of spores in the isolated strains, saturated grown cultures underwent a heat treatment at 100 °C for 30 min to eliminate vegetative cells [32]. Subsequently, 100 μL of the boiled culture was inoculated in fresh liquid ATM1 medium, and the cultures were incubated at 60 °C for 48 h. Each strain was cultured in triplicate, and a blank control without any inoculum and a growth control (bacteria without temperature treatment) for each evaluated strain were included. The growth of the cultures was assessed by measuring the optical density (O.D) at 600 nm, following the previously described methodology.

### 2.6. DNA Extraction

For DNA extraction, 3 mL of fresh bacterial culture were centrifuged, and the resulting pellet was processed using the Wizard^®^ Genomic DNA Purification Kit (Promega, Madison, WI, USA), following the manufacturer’s instructions. To assess the integrity of the DNA, horizontal electrophoresis was performed on a 0.5% agarose gel containing ethidium bromide in 0.5xTAE buffer. The DNA was fractionated at room temperature and visualized on a UV transilluminator (Vilver Lourmat, B.P 66 TORCY -Z.I. SUD 77202, Marne La Vallee, Cedex 1 France).

### 2.7. Taxonomic Identification of the Isolates

The 16S rRNA gene was amplified from the extracted DNA by polymerase chain reaction (PCR) using the Eubacterial primers 27F (5′-AGAGTTTGATCMTGGCTCAG-3′) and 1525R (5′-AAGGAGGTGWTCCAGCC-3′) [33], using an MJ Research PT-100 Thermal Cycler (MJ Research, Inc., Waltham, MA, USA). The amplicons were cloned in pGEM-T easy vector (Promega, Madison, WI, USA), and the insert was sequenced (Macrogen—Korea, Seoul, Republic of Korea). Nucleotide sequences were analyzed for similarity to the 16S rRNA gene using the BLAST program [34]. The sequence was also aligned with those of the reference strains using CLUSTAL W [35]. The evolutionary history was inferred using the Maximum Likelihood method based on the Jukes–Cantor model [36]. The tree with the highest log likelihood (−2281.3475) is shown. The percentage of trees in which the associated taxa clustered together is shown next to the branches. Initial tree(s) for the heuristic search were obtained automatically by applying Neighbor-Join and BioNJ algorithms to a matrix of pairwise distances estimated using the Maximum Composite Likelihood (MCL) approach, and then selecting the topology with the superior log likelihood value. The tree was drawn to scale, with branch lengths measured in the number of substitutions per site. The analysis involved 44 nucleotide sequences. All positions containing gaps and missing data were eliminated. There were a total of 642 positions in the final dataset. Evolutionary analyses were conducted in MEGA6 [37].

**Table 1 microorganisms-12-00473-t001:** Composition of the culture media used in this study.

Culture Medium	Organic Source	Mineral Salts	Reference
TT medium	0.4% Yeast extract 0.8% Peptone	0.2% (*w*/*v*) NaCl	[38]
YPS	0.1% Yeast extract 0.4% Peptone	0.5% Artificial seawater	[39]
Miller	2 g/L Yeast extract	1.5 g/L KCl 0.5 g/L K_2_HPO_4_ 0.5 g/L KH_2_PO_4_	[40]
Kunaemneni	Solution A: 0.75% Glucose 0.5% Peptone	Solution B: 0.5% (*w*/*v*) MgSO_4_ 0.5% (*w*/*v*) KH_2_PO_4_ 0.01% (*w*/*v*) FeSO_4_. 7 H_2_O Mix 0.5% *v*/*v* of solution A in solution B.	[41]
Huang	0.5% Yeast extract 1% Peptone 0.5 g/L Glucose	0.4 g/L Na_2_HPO_4_ 0.085% (*w*/*v*) Na_2_CO_3_ 0.02 g/L ZnSO_4_ 0.02 g/L MgSO_4_ 0.02 g/L CaCl_2_	[42]
ATM 1	0.3% Yeast extract 0.25% Peptone	AMT salt solution: 832.42 mg/L CaCl_2_ 8.22 mg/L MgCl_2_ 334.85 mg/L NaCl 4697.4 mg/L KCl 16.763 mg/L SiO_2_ 77.832 mg/L NaHSO_4_ 14.377 mg/L CsCl 0.030 mg/L FeSO_4_ 0.025 mg/L CuSO_4_	Designed based on the results of Fernandez-Turiel et al. (2005) [28,43].
ATM 2	0.1% Yeast extract 0.4% peptone	ATM salt solution: Same as ATM1.	Designed based on the results of Fernandez-Turiel et al. (2005) [28,43].

### 2.8. Depositions of Strains

All the strains isolated during this study were deposited in the Microbial Stock at the Extremophile Laboratory, located in the Antofagasta Institute at the University of Antofagasta, Chile.

## 3. Results

### 3.1. Sample Characteristics

The samples were taken from five different ecosystems distributed in the Northern Macrozone of Chile, as shown in the map (Figure 1). The temperature and pH ranges of the water samples or sludges in the different sampling sites were between 22 and 70 °C and pH 6.5–9.1, respectively (Table 2).

### 3.2. Cultivation and Isolation of Thermophilic Microorganisms

The Natural Tatio Media (NTM) enabled the isolation of 15 thermophilic bacteria, all presenting colonies with similar characteristics. The evaluation of different culture media, intended for the development of a laboratory cultivation medium, led to the successful cultivation of thermophilic microorganisms. Notably, the ATM1 medium demonstrated vigorous proliferation for the analyzed samples (Table 3). Additionally, certain strains exhibited proliferation in Miller and Huang media, albeit to a lesser extent compared to ATM1 media.

The cultivation results indicate that, overall, the selected strains exhibited proliferation within a temperature range of 40 to 70 °C. This characteristic classifies the isolates as strict thermophiles [44]. The optimal pH range for proliferation is generally found between 6 and 9, with a dominant pH optimum between 6 and 7. Interestingly, the pH values closely resemble those found in the sampling areas (see Table 2). Observations under an optical microscope using Gram staining showed that all isolates corresponded to Gram-positive bacilli (see Table 3). The photographs of the Gram stains of the fifteen bacteria are shown in the Appendix A. The sporulation tests for the 15 isolated strains showed that all the strains were able to sporulate.

### 3.3. Taxonomic Identification Based on 16S rRNA Sequencing and Phylogenetic Analysis

DNA analyses of the 15 strains isolated in this study have indicated that the strains belong to three genera: *Anoxybacillus* sp. with 8 isolates, *Geobacillus* sp. with 2 isolates, *Parageobacillus* with 1 isolated and *Aeribacillus* sp. with 4 isolates. Phylogenetic relationships with closely related organisms are shown on the phylogenetic tree (Figure 2). All of them exhibit high similarity to previously described species (>98% sequence similarity) isolated from other locations (Table 4). Furthermore, the results for strains M3 and Q1 are noteworthy, showing a high similarity to *Aeribacillus* strains (100%), described as isolated strains [45,46].

### 3.4. Determination of Hydrolytic Enzymes Activities in Isolated Bacteria

In order to assess the abilities of the isolated strains to produce enzymes of biotechnological interest, the hydrolysis of proteinaceous, lipidic, cellulolytic, and amylolytic substrates was evaluated in culture. The identification of hydrolases revealed that the most frequent type consisted of proteases, followed by amylases and lipases in a lower percentage. No cellulases were detected using the methodology described in the Material and Methods section. Notably, strain 702B exhibited all three types of hydrolytic activity. The results of the hydrolytic activities can be found in Table 5.

## 4. Discussion

The northern region of Chile spans a broad altitudinal and latitudinal spectrum, hosting diverse aquatic ecosystems and giving rise to various landscapes and geological features, creating polyextreme habitats for microorganisms [47]. Hot springs provide unique environments for thermophilic microorganisms, characterized by elevated temperatures and often limited nutrient levels [48,49]. However, thermophilic microorganisms are not exclusive to geothermal environments, as they have been discovered in hay [50], deep-subsurface oil reservoirs [51], sugar refinery wastewater [52], hypersaline lakes [53], and cool soil environments [54,55,56]. In this study, we isolated fifteen axenic thermophilic bacteria from two geothermal systems in the high plains of the Andes, two hypersaline lagoons in the heart of the Atacama Desert, and a freshwater effluent near the coastal area in the city of Antofagasta. These findings align with previous studies, indicating the presence of thermophilic and extremophilic microorganisms in a wide range of environments.

The isolated microorganisms exhibited a phylogenetic affiliation with the genera *Anoxybacillus*, *Aeribacillus*, *Parageobacillus*, and *Geobacillus*. These genera belong to the family *Bacillaceae*, order *Bacillales*, and phylum *Firmicutes*; they are recognized for their ability to form endospores and are commonly found in various soil environments [1]. It is important to note that some members of the phylum *Firmicutes* are industrially valuable for the development of different biotechnological products, like antibiotics, enzymes, and dairy products [57,58]. In this research, the demonstrated ability of these microorganisms to produce hydrolytic enzymes highlights the significance of further characterizing these isolates for their biotechnological use.

The genus *Anoxybacillus*, initially characterized by Pikuta in 2000 [59], which originally consisted of 2 species, *Anoxybacillus pushchinensis* and *Anoxybacillus flavithermus*, has expanded to encompass 24 species and two subspecies as of 2023 [60]. This genus comprises thermophilic and alkaliphilic microorganisms and has been isolated from diverse locations worldwide [61,62]. In this work, we isolated microorganisms of this genus—one isolate in Jurasi hot springs and seven isolates at El Tatio—with high similarity (99.2 to 99.9%) to species of *kamchatkensis*, *tepidamans*, *kualawohkensis*, and *flavithermus*. Interestingly, the 16S rRNA gene sequences of the genus *Anoxybacillus* show a close phylogenetically group of bacteria that might not allow strains to be identified up to the species level [63,64]. The microorganisms of this genus are predominant in thermophilic environments, containing several species and strains that could be differentiated among them by Amplified Ribosomal DNA Restriction Analysis (ARDRA) and phenotypic characteristics [64]. Similarly, the *Anoxybacillus* isolates of this study do not differ much in terms of microscopic observations, temperature, and pH conditions, and are similar to previous reports of *Anoxybacillus* characterizations [62,63,64,65,66,67].

The significant representation of members from the genus *Anoxybacillus* spp. among the total number isolates holds immense promise for prospective biotechnological inquiries. This genus has been associated with a spectrum of valuable proteins, including amylolytic enzymes [68], xylose isomerases [69], thermostable endonucleases [70], diphenolases [71], and fructose-1,6-bisphosphate aldolases [72]. This abundance of potential applications underscores the significance of further exploring the biotechnological capacities inherent in *Anoxybacillus* spp. [73]. In this context, several *Anoxybacillus* isolates from the mentioned environment exhibit notable hydrolytic activities, particularly amylase in strains 2a55, 702B, SA, M8, and 3A55 (in ETGF), as well as C3 in the Cejar lagoon, and M3 and Q1 in Quebrada Carrizo. Additionally, certain strains, including 702B, 2A55, SA, M8, 3A55 (ETGF), C3 (Cejar lagoon), M3 and Q1 (Quebrada Carrizo), display proteolytic activities. Lipolytic activity was observed in only three strains, namely, 702B, 2A55 (ETGF), and Q1 (Quebrada Carrizo). Notably, isolates 702B and 3A55 exhibited positive results for all three enzymatic activities defined earlier. All strains tested negative for cellulase activity.

Strains CT1, 2B55, and 2**55* were negative for all four hydrolytic activities studied. Furthermore, the phylogenetic proximity of these strains to *Anoxybacillus kualawohkensis* strain ET103, as determined by the 16S rRNA gene sequence, showed high similarity (99.5%, 98.4%, and 99.4%, respectively). The optimal pH values for culture conditions are very similar, ranging from 6 to 8 with an optimum at 8. The strains present slight differences in the temperature for optimal growth, with CT1 ranging between 40 and 65 °C and optimal at 60 °C, 2*55 being within the same range, but with an optimum at 65 °C, and 2B55 exhibiting growth between 50 to 65 °C and optimal at 60 °C. These differences suggest that despite their close phylogenetic relationship exposed by 16S rRNA analysis, these strains may have acquired adaptations to proliferate at different temperature ranges.

The genus *Geobacillus* was documented initially by Nazina [74], and initially ten species were identified [75]. This genus is composed of thermoresistant (25–45 °C) and obligate thermophilic microorganisms (50 °C to 75 °C), proliferating in a pH range between 6.0 and 8.5, that are chemo-organotrophic, including aerobic, anaerobic, and facultative anaerobic members, classifications consistent with those observed in the characterization of the strains isolated in this study [76].

Through whole-genome sequencing, researchers have identified two distinct clades within the *Geobacillus* genera, characterized by variations in guanine + cytosine contents and nucleotide base composition. This observation led to the proposal of a new genus, *Parageobacillus*, specifically for *Geobacillus* species within clade II, establishing it as a novel genus. Consequently, *Geobacillus thermoglucosidasius* is now classified as *Parageobacillus thermoglucosidasius* in this revised taxonomy [77].

The genus *Aeribacillus*, a reclassification of *Geobacillus pallidus,* distinguishes itself from *Anoxybacillus* and *Geobacillus* through variations in DNA G+C content, as well as fatty acid and polar lipid profiles. Its hallmark features include anaerobic C16:0 major fatty acid(s) and a DNA G+C content of 39–41 mol%, setting it apart from *Anoxybacillus* and *Geobacillus*, which exhibit higher DNA G+C contents (42–57 and 48–58, respectively) and function as facultative anaerobes [78]. Various strains of *Aeribacillus pallidus* have been isolated from diverse environments, such as sewage, oil reservoirs, hot springs, oil-contaminated soil, and deep geothermal reservoirs [78]. In this study, we identified four strains from this genus: CJ1 in Jurasi, TB4 in Tebenquiche Lagoon, and M3 and Q1 in Quebrada Carrizo. The initial three isolates exhibited a significant resemblance to *Aeribacillus pallidus* P18 (APA022323), with similarity percentages of 99.9%, 99.9%, and 100%, respectively (45). While the fourth isolate shows 100% similarity with *Aeribacillus pallidus* strain 21KAM21, a bacterium isolated from uncontaminated environmental sites in Milos, Greece [46]. 

This study emphasizes the biotechnological richness of thermophilic microorganisms in northern Chile, providing opportunities for future research and applications in various fields, from industrial enzyme production.

## 5. Conclusions

The development of this study has culminated in the establishment of a collection comprising axenic thermophilic strains extracted from aquatic ecosystems in northern Chile. The distinct environmental characteristics of these ecosystems have endowed us with a wealth of genetic resources from extremophilic microorganisms with great biotechnological potential. Representing the phylum *Firmicutes*, strains within the *Anoxybacillus*, *Aeribacillus*, *Geobacillus*, and *Parageobacillus* genera, situated within the bacteria domain, provide a robust genetic platform poised for in-depth exploration in the fields of thermally stable enzymes, recombinant production, and industrial applications. Among the fifteen thermophilic microorganisms within this collection, notable hydrolytic enzymatic activities—amylolytic, proteolytic, and lipolytic—have been identified as predominant. These enzymatic traits highlight the diversity within this unique assembly of thermophiles, but also emphasize their potential use as a thermophilic chassis for synthetic biology.

## Figures and Tables

**Figure 1 microorganisms-12-00473-f001:**
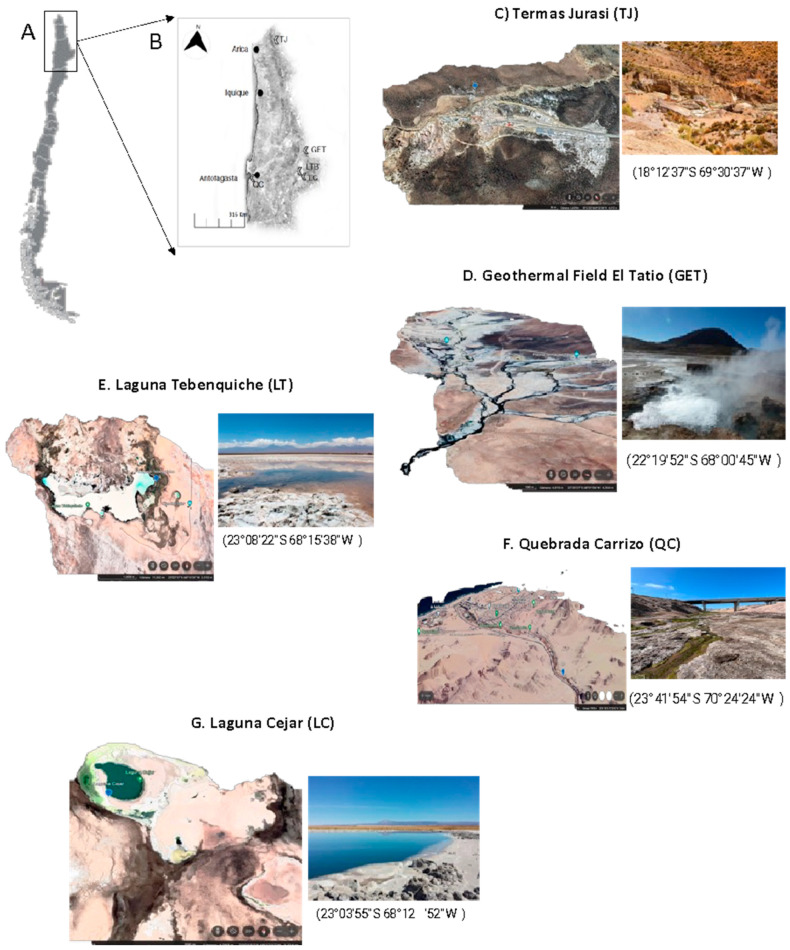
Sampling sites. (**A**) Map of Chile. (**B**) North macrozone of Chile, the main cities (Arica, Iquique and Antofagasta) are indicated with black spheres. The sampling points are indicated with open arrows. (**C**) Termas Jurasi (TJ). (**D**) El Tatio Geothermal Field (GET). (**E**) Laguna Tebenquiche. (**F**) Quebrada Carrizo. (**G**) Laguna Cejar.

**Figure 2 microorganisms-12-00473-f002:**
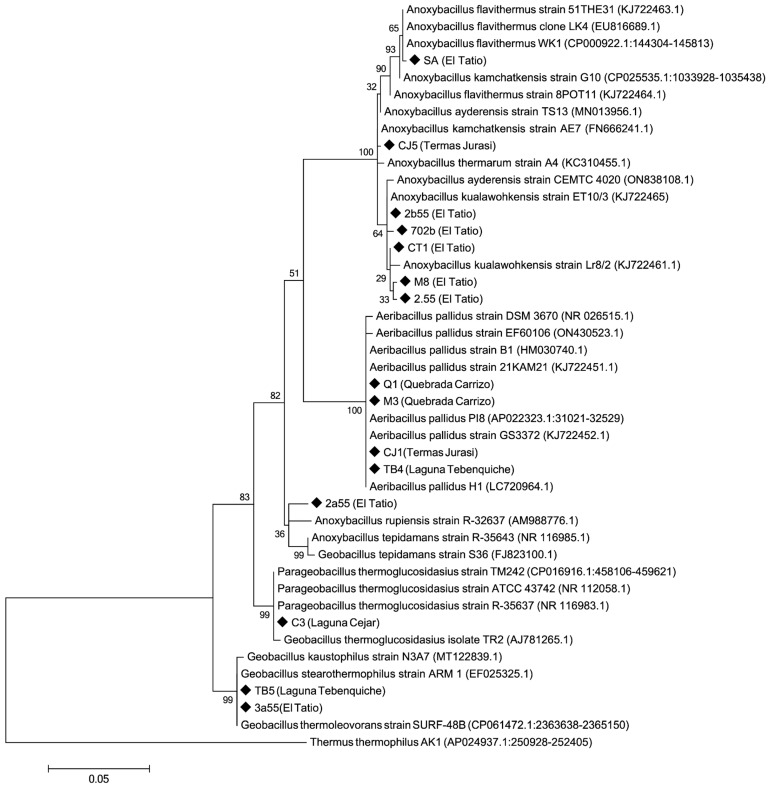
Phylogenetic tree of 16S rRNA gene sequences of isolates and closely related bacteria. The evolutionary history was inferred using the Maximum Likehood method and Jukes–Cantor model. The percentages of replicate trees in which the associated taxa clustered together in the bootstrap test (1000 replicates) are shown next to the branches. *Thermus thermophilus* AK1 (AAP024937) was used as an outgroup.

**Table 2 microorganisms-12-00473-t002:** Environmental characteristics of the sampling sites.

Site	Temperature (°C)	pH	Classification	Location
Termas Jurasi	48–52	7.5–8.2	Thermal affluent	18°12′37″ S 69°30′37″ W 4053 m
El Tatio Geothermal Field	55–82	7.5–8.2	Geothermal field	22°19′52″ S 68°00′45″ W 4278 m
Laguna Tebenquiche	12–22	7.8–8.2	Hypersaline lagoon, Atacama Desert	23°08′22″ S 68°15′38″ W 2316 m
Laguna Cejar	22–32	7.4–8.6	Hypersaline lagoon, Atacama Desert	23°03′55″ S 68°12′52″ W 2334 m
Quebrada Carrizo	18–22	6.8–7	Fresh water spring, Coastal Desert	23°41′54″ S 70°24′24″ W 95 m

**Table 3 microorganisms-12-00473-t003:** Cultivation parameters of isolated thermophilic bacteria.

Source	Iso-late	Media							Temperature Range/Optimal (°C)	pH Range/Optimal	Morphology	
		TT	YPS	Miller	Kunaemnemi	Huang	ATM1	ATM2			Gram	Colony
Termas Jurasi (2 isolated)	CJ1	-	-	++	-	++	+++	+	55–70/65	6–8/7	Positives Bacillaries	Whitish, edges uniform, convex surface
	CJ5	-	-	+	-	-	+++	-	45–60/60	6–9/8	Positives Bacillaries	Whitish, edges uniform, convex surface
-El Tatio Geothermal Field (9 isolated)	CT1	-	-	-	-	-	+++	-	40–65/55	6–9/8	Positives Bacillaries	Whitish, edges uniform, convex surface
	702B	-	-	+	-	+	+++	-	50–70/60	6–9/8	Positives Bacillaries	Whitish, edges uniform, convex surface
	2A55	-	-	-	-	+	+++	-	50–65/55	6–9/7	Positives Bacillaries	Whitish, edges uniform, convex surface
	2B55	-	-	+	-	+	+++	-	50–65/60	6–9/8	Positives Bacillaries	Whitish, edges uniform, convex surface
	SA	-	-	+	-	++	+++	+	45–70/55	6–8/6	Positives Bacillaries	Whitish, edges uniform, convex surface
	2*55	-	-	+	-	+	+++	-	45–65/60	6–8/8	Positives Bacillaries	Whitish, edges uniform, convex surface
	M8	-	-	+	-	+	+++	-	50–70/50	6–8/8	Positives Bacillaries	Whitish, edges uniform, convex surface
	3A55	-	-	+	-	+	+++	+	40–65/55	6–8/7	Positives Bacillaries	Whitish, edges uniform, convex surface
Laguna Tebenquiche (2 isolated)	TB4	-	-	+	-	+	+++	+	50–65/55	6–8/8	Positives Bacillaries	Whitish, edges uniform, convex surface
	TB5	-	-	+	-	+	+++	+	45–60/55	6–8/7	Positives Bacillaries	Whitish, edges uniform, convex surface
Laguna Cejar (1 isolated)	C3	-	-	+	-	+	+++	+	45–65/60	6–8/8	Positives Bacillaries	Whitish, edges uniform, convex surface
Quebrada Carrizo (2 isolated)	M3	-	-	++	-	+	+++	+	45–60/50	6–9/7	Positives Bacillaries	whitish, edges sawn, surface convex
	Q1	-	-	+	-	+	+++	-	45–70/60	6–9/8	Positives Bacillaries	Whitish, edges uniform, convex surface

Growth scale: -, not colony growth; +, low optical density of cell; ++, good growth; +++, the best growth. Gram staining is shown in Appendix A.

**Table 4 microorganisms-12-00473-t004:** Taxonomic identification of the isolates based on 16S rRNA gene sequences.

Source	Isolate	Closest Strain	Similarity (%)	Deposited Number
Termas Jurasi (2 isolated)	CJ1	*Aeribacillus pallidus* PI8	99.9	AP022323.1
	CJ5	*Anoxybacillus kamchatkensis* strain G10	99.6	CP025535.1
El Tatio Geothermal Field (8 isolated)	CT1	*Anoxybacillus kualawohkensis* strain ET103	99.5	KJ722465.1
	702B	*Anoxybacillus kualawohkensis* strain ET103	99.2	KJ722465.1
	2A55	*Anoxybacillus tepidamans* strain R-35643	98.4	NR_116985.1
	2B55	*Anoxybacillus kualawohkensis* strain ET103	98.4	KJ722465.1
	SA	*Anoxybacillus flavithermus* clone LK4	99.8	EU816689.1
	2*55	*Anoxybacillus kualawohkensis* strain ET103	99.4	KJ722465.1
	M8	*Anoxybacillus kualawohkensis* strain ET103	99.2	KJ722465.1
	3A55	*Geobacillus thermoleovorans* strain SURF-48B	99.9	CP061472.1
Laguna Tebenquiche (2 isolated)	TB4	*Aeribacillus pallidus* PI8	99.8	AP022323.1
	TB5	*Geobacillus thermoleovorans* strain SURF-48B	99.8	CP061472.1
Laguna Cejar (1 isolated)	C3	*Parageobacillus thermoglucosidasius* strain TM242	99.8	CP016916.1
Quebrada Carrizo (2 isolated)	M3	*Aeribacillus pallidus* PI8	100	AP022323.1
	Q1	*Aeribacillus pallidus* strain 21KAM21	100	KJ722451.1

**Table 5 microorganisms-12-00473-t005:** Hydrolase activity produced by thermophilic isolated bacteria.

Source	Isolated Strain	Closest Strain	Protease Activity	Amylase Activity	Lipase Activity	Cellulase Activity
Termas Jurasi (2 isolated)	CJ1	*Aeribacillus pallidu*s PI8	++	-	-	-
	CJ5	*Anoxybacillus kamchatkensis* strain G10	+	-	-	-
El Tatio Geothermal Field (8 isolated)	CT1	*Anoxybacillus kualawohkensis* strain ET103	-	-	-	-
	702B	*Anoxybacillus kualawohkensis* strain ET103	++	+	+	-
	2A55	*Anoxybacillus tepidamans* strain R-35643	-	++	-	-
	2B55	*Anoxybacillus kualawohkensis* strain ET103	-	-	-	-
	SA	*Anoxybacillus flavithermus* clone LK4	-	+++	-	-
	2*55	*Anoxybacillus kualawohkensis* strain ET103	-	-	-	-
	M8	*Anoxybacillus kualawohkensis* strain ET103	+	+	-	-
	3A55	*Geobacillus thermoleovorans* strain SURF-48B	+	++	+++	-
Laguna Tebenquiche (2 isolated)	TB4	*Aeribacillus pallidus* PI8	+	-	-	-
	TB5	*Geobacillus thermoleovorans* strain SURF-48B	++	-	-	-
Laguna Cejar (1 isolated)	C3	*Parageobacillus thermoglucosidasius* strain TM242	++	+++	-	-
Quebrada Carrizo (2 isolated)	M3	*Aeribacillus pallidus* PI8	-	+	-	-
	Q1	*Aeribacillus pallidus* strain 21KAM21	-	+	+	-

-, enzymatic activity not detected; +, low enzymatic activity detected; ++, high enzymatic activity detected; +++, the best enzymatic activity detected.

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
