# Peer review of "Isolation of Thermophilic Bacteria from Extreme Environments in Northern Chile"

_microorganisms, 2024, doi:10.3390/microorganisms12030473_

Round 1

Reviewer 1 Report

Comments and Suggestions for Authors

The study is devoted to isolation of thermophilic microorganisms with hydrolytic activities inhabiting extreme environments from Northern Chile. The authors isolated 15 strains of thermophilic bacteria. According to phylogenetic analysis based on 16S rRNA gene these strains belong to genera Parageobacillus, Geobacillus, Anoxybacillus, and Aeribacillus. Moreover presence of amylolytic, cellulolytic, lipolytic and proteolytic activities was tested. This work is quite interesting because not many articles about thermophiles from Chile are known, but several points must be improved:

-one of main problem is absence of clear focus of the study. The research went in several directions but not in depth. For example, 1) the authors isolated strains but the strains were described very briefly (there were no substrate spectra, no comparison with known species), 2) hydrolytic activities were determined only using plate method however there are more advance methods (e.g. zymography), 3) no genome sequencing was performed but it is useful both for phylogenomic analysis and for identification of the genes encoding target enzymes.

-why did the authors use yeast extract, peptone or glucose for isolation os pure cultures if the authors planned to study hydrolytic microorganisms? Why were  respective substrates (e.g. starch, cellulose, some proteins and lipids) not used?

-Discussion section should be improved and should contain only material closely related to goal of the study but not generalized aspects. For example, at L328 the authors discuss divalent metal adsorption and Cd adsorption by members of Anoxybcillus. At L362-365 and L381-384 ethanol and bacteriocins production are discussed. Why? How are these moments related to the current study? If the authors propose that isolated strains possess abovementioned properties, they should confirmed it exprerimentally.

Minor comments:

L150: why was "4', 6-diamidino-2-phenylindole" given in quotes?

L158: did the authors mean three culture transfers following each other?

L160: the sentence at LL162-164 look like a repeat sentence at LL160-161

L168: why was method of cellulolytic activity detection not described?

L254: please provide figures with cell morphology and spores

L261: actually there are different bacterial species with >98.7% identity of 16S rRNA gene sequences. For accurate species assignment it is necessary to perform whole genome comparisons (DNA-DNA hybridization, ANI).

L269: please specify that the tree was based on 16S rRNA genes comparison. Why did the authors use Neighbor-Joining but not Maximum Likelihood method? Please provide tree with branches showing evolutionaty distances between novel isolates and known species.

L283: please replace “Amilase acivity” with “Amylase activity”

L319: what is ARDRA?

L394: please replace “extremophile microorganisms” with “extremophilic microorganisms” or “extremophiles”

Author Response

Reviewer 1

-One of main problem is absence of clear focus of the study. The research went in several directions but not in depth. For example, 1) the authors isolated strains but the strains were described very briefly (there were no substrate spectra, no comparison with known species), 2) hydrolytic activities were determined only using plate method however there are more advance methods (e.g. zymography), 3) no genome sequencing was performed but it is useful both for phylogenomic analysis and for identification of the genes encoding target enzymes. 

We appreciate the comments and suggestions provided by this reviewer, especially regarding the examples suggested in points 1, 2, and 3. These examples present a valuable opportunity to achieve a deeper characterization of our work. It is important to note that our publication focuses on efforts dedicated to the isolation of thermophilic microorganisms and their subsequent cultivation under laboratory conditions.

This approach provides us with the opportunity to conduct more detailed molecular characterizations in future stages. The availability of these microorganisms will facilitate our research group in carrying out more comprehensive investigations. Among the upcoming phases of our study, we plan to include the genome sequencing of the microorganisms isolated, as well as deeper biochemical characterizations of the identified enzymes. Additionally, we are working to explore the usage characteristics and substrate fermentation by these microorganisms.

These elements are integral to our ongoing research, and we look forward to the opportunity to share the results obtained in future publications. We appreciate the reviewer's interest and attention to key areas of our work, and we hope that these clarifications strengthen the understanding of the direction and scope of our project.

-why did the authors use yeast extract, peptone or glucose for isolation os pure cultures if the authors planned to study hydrolytic microorganisms? Why were respective substrates (e.g. starch, cellulose, some proteins and lipids) not used?

This is an excellent observation by the reviewer, and we fully agree at this stage of reading and revision. Initially, the idea of cultivating thermophiles did not specifically consider isolating organisms with hydrolytic activities. However, while obtaining the results during the microbiological characterization, it became evident that these microorganisms exhibited the ability to hydrolyze several substrates. For this reason, we decided to incorporate this information into the title. To address this point, we have chosen to modify the title of this article to more accurately reflect the aim for the isolation of thermophilic organisms.

-Discussion section should be improved and should contain only material closely related to goal of the study but not generalized aspects. For example, at L328 the authors discuss divalent metal adsorption and Cd adsorption by members of Anoxybacillus. At L362-365 and L381-384 ethanol and bacteriocins production are discussed. Why? How are these moments related to the current study? If the authors propose that isolated strains possess abovementioned properties, they should confirmed it exprerimentally.

The reviewer is absolutely correct with these suggestions. We became overly enthusiastic about comparing features with organisms of known species and delving into aspects that are still under investigation, such as ethanol production. For instance, we are currently conducting research on one of the bacteria reported in this publication, characterizing its ability to produce ethanol. Despite this ongoing work, for this particular study that is focused on the isolation and general characterization, highlighting  aspects related to laboratory cultivation, it doesn't make sense to include such details in the discussion without presenting experimental evidence. To address this issue, we have decided to remove these elements from the discussion section.

Minor comments:

L150: why was "4',6-diamidino-2-phenylindole" given in quotes?

 This corresponds to a typing error, there is no particular reason. It was corrected in the text.

L158: did the authors mean three culture transfers following each other?

We appreciate being informed of this clarification; it is prone to confusion as currently written. What we intended to convey is that each evaluated pH point examined had three culture tubes as replicas, including three replicas of the control without inoculum for each pH point. This has been modified in the text.

L160: the sentence at LL162-164 look like a repeat sentence at LL160-161

Indeed, this corresponds to a repetition that was not edited correctly. It has been corrected in the text.

L168: why was method of cellulolytic activity detection not described?

We appreciated this observation by the reviewer, we regret having forgotten to include this method in the methodology section. It has been added in section 2.4.5.

L254: please provide figures with cell morphology and spores

We have included a supplementary figure featuring the Gram staining of each strain. In the case of spore images, it is not feasible to provide them as our methodology did not involve obtaining such images. The process begins with the understanding that the cells of these bacteria do not remain viable after 30 minutes of boiling at 100°C, leaving only intact spores. When placed in a new medium under suitable conditions for cultivation, these spores germinate and regrow.

L261: actually there are different bacterial species with >98.7% identity of 16S rRNA gene sequences. For accurate species assignment it is necessary to perform whole genome comparisons (DNA-DNA hybridization, ANI).

We agree with the reviewer. The accurate identification of these species would require whole-genome comparisons using methods such as DNA-DNA hybridization or ANI (Average Nucleotide Identity). In our study, we focused on obtaining isolates of thermophiles. In subsequent studies, with these isolates available in the laboratory, we are planning to achieve a more in-depth characterization of these strains.

L269: please specify that the tree was based on 16S rRNA genes comparison. Why did the authors use Neighbor-Joining but not Maximum Likelihood method? Please provide tree with branches showing evolutionaty distances between novel isolates and known species.

The NJ method was generated using 16S rRNA genes for comparison. While the maximum likelihood tree is more flexible in terms of parametrization, we use the NJ method for a rapid and preliminary exploratory analysis that shows the clade that groups these strains.

We took the reviewer's suggestion and constructed a ML tree method using the Jukes-Cantor model was used, assuming that all nucleotide substitution rates are equal, all bases are equally probable at all positions in the sequence, and it does not take into account the heterogeneity of substitution rates or the presence of invariant sites.

The new tree shows the evolutionary distances of the branches of the new isolates, while maintaining the clade associations of the strains.

L283: please replace “Amilase acivity” with “Amylase activity”

We thank the reviewer for bringing attention to this misspelling; it has been corrected in the text.

L319: What is ARDRA?

Amplified Ribosomal DNA Restriction Analysis (ARDRA) is a technique that uses three restriction enzymes (Alu I, Msp I, and Hae III) to cluster sequences of microorganisms with high similarity index. This tool is employed for identification through the analysis of 16S rRNA gene sequences, followed by cleavage with restriction enzymes and phylogenetic classification by assigning them to clusters. The meaning of the acronym was incorporated into the text.

L394: please replace “extremophile microorganisms” with “extremophilic microorganisms” or “extremophiles”

It has been corrected in the text and replaced for extremophilic microorganisms. 

Reviewer 2 Report

Comments and Suggestions for Authors

In this study, 15 strains of thermophilic bacteria were isolated from the extreme environment of northern Chile, and these thermophiles exhibited significant hydrolase activity. The purpose of the experiment is clear, but the innovation needs to be improved. The specific questions in this article are as follows:

1. For Natural Tatio Media (NTM) medium ingredients, what is the volume of El Tatio hot spring water sample required for adding 0.25g yeast extract and 0.20g peptone? How to determine the amount of yeast extract and peptone?

2. In the description of the separation method for thermophilic bacteria, it is recommended to indicate the time of static and dynamic cultivation of the obtained bacterial culture.

3. Many contents of this article do not match the charts, for example, the table corresponding to the temperature and pH range of the water sample or sludge at the sampling point is wrong, it is recommended to check carefully.

4. The optimal growth temperature for bacteria is obtained by measuring the absorbance value of bacterial suspension at 600nm. Why is the absorbance value of 660nm used in the determination of spore formation??

5. In this article, researchers focus on the industrial application potential of enzymes derived from thermophilic bacteria. Does the activity of hydrolytic enzymes derived from thermophilic bacteria increase compared to enzymes derived from mesophilic bacteria? How to further obtain and evaluate the potential application of hydrolytic enzymes derived from isolated thermophilic bacteria?

6. What is the basis for determining the optimal culture temperature and temperature growth range of these isolated thermophilic bacteria in Table 3?

7. Does the selection of different buffer solutions have an impact on the optimal pH determination of isolated thermophilic bacteria?

Comments on the Quality of English Language

In this study, 15 strains of thermophilic bacteria were isolated from the extreme environment of northern Chile, and these thermophiles exhibited significant hydrolase activity. The purpose of the experiment is clear, but the innovation needs to be improved. The specific questions in this article are as follows:

1. For Natural Tatio Media (NTM) medium ingredients, what is the volume of El Tatio hot spring water sample required for adding 0.25g yeast extract and 0.20g peptone? How to determine the amount of yeast extract and peptone?

2. In the description of the separation method for thermophilic bacteria, it is recommended to indicate the time of static and dynamic cultivation of the obtained bacterial culture.

3. Many contents of this article do not match the charts, for example, the table corresponding to the temperature and pH range of the water sample or sludge at the sampling point is wrong, it is recommended to check carefully.

4. The optimal growth temperature for bacteria is obtained by measuring the absorbance value of bacterial suspension at 600nm. Why is the absorbance value of 660nm used in the determination of spore formation??

5. In this article, researchers focus on the industrial application potential of enzymes derived from thermophilic bacteria. Does the activity of hydrolytic enzymes derived from thermophilic bacteria increase compared to enzymes derived from mesophilic bacteria? How to further obtain and evaluate the potential application of hydrolytic enzymes derived from isolated thermophilic bacteria?

6. What is the basis for determining the optimal culture temperature and temperature growth range of these isolated thermophilic bacteria in Table 3?

7. Does the selection of different buffer solutions have an impact on the optimal pH determination of isolated thermophilic bacteria?

Author Response

Reviewer 2

  1. For Natural Tatio Media (NTM) medium ingredients, what is the volume of El Tatio hot spring water sample required for adding 0.25g yeast extract and 0.20g peptone? How to determine the amount of yeast extract and peptone?

We appreciated the reviewer observation, we did not realize that the unit of weight/volume was missing. The text has been modified as follows: we employed Natural Tatio Media (NTM), composed of a water sample from the El Tatio thermal spring (filtered through a 0.2 μm environmental water filter) and supplemented with 0.25 % (w/v) of yeast extract and 0.20 % (w/v) of peptone.

In our aim to cultivate thermophilic microorganisms, we considered that the quantity of organic matter in thermal ecosystems is low. Additionally, some thermophilic organisms do not tolerate high substrate concentrations. We referred to the cultivation media of Thermus, as reported by T. Brock, indicating that Thermus aquaticus can grow up to 0.3% p/v tryptone and yeast extract but not beyond. In contrast, another species, T. thermophilus (Oshima and Imahori, 1974), can grow in concentrations of organic constituents up to 1% p/v (https://doi.org/10.1007/978-1-4612-6284-8_4). In our initial thermophile cultivation experiences, we used similar concentrations between 0.1 and 1% of both at similar concentrations. The concentration indicated in this publication corresponds to the one that yielded the best results in the necessary time to achieve growth, observed by suitable turbidity of the media.

  1. In the description of the separation method for thermophilic bacteria, it is recommended to indicate the time of static and dynamic cultivation of the obtained bacterial culture.

It was incorporated into the text as follows: The initial culture temperature was maintained at 60°C for static cultures for 4 days, using a SHEL LAB incubator (model EI2-2, Sheldon Manufacturing, INC 300 N. 26TH Cornelius, OR 97113, USA); while shaking cultures at 60°C for 4 days utilized a LAB NET shaker incubator (LabNet 311DS, Marshall Scientific, Co., Huntington, WV, USA) operating at 250 rpm.

  1. Many contents of this article do not match the charts, for example, the table corresponding to the temperature and pH range of the water sample or sludge at the sampling point is wrong, it is recommended to check carefully.

Certainly, an error in the results description (3.2) was identified, as it incorrectly referenced Table 2 for the culture media where the bacteria grew, when it should have been Table 3. This has been rectified. Additionally, in section 3.1, there was a reference to Table 1 for the pH and temperature of the samples; however, the accurate reference should be Table 2. This correction has been made in the text.

Something similar in: Gram staining showed that all isolates corresponded to Gram-positive bacilli (see Table 2). This was changed to Table 3.

A similar situation occurred in: "All of them exhibit high similarity with previously described species (>98% sequence similarity) isolated from other locations (Table 3)." This was changed to Table 4.

  1. The optimal growth temperature for bacteria is obtained by measuring the absorbance value of bacterial suspension at 600nm. Why is the absorbance value of 660nm used in the determination of spore formation??

This is a typing error, it was corrected in the text to 600 nm.

  1. In this article, researchers focus on the industrial application potential of enzymes derived from thermophilic bacteria. Does the activity of hydrolytic enzymes derived from thermophilic bacteria increase compared to enzymes derived from mesophilic bacteria? How to further obtain and evaluate the potential application of hydrolytic enzymes derived from isolated thermophilic bacteria?

Does the activity of hydrolytic enzymes derived from thermophilic bacteria increase compared to enzymes derived from mesophilic bacteria?

Generally, thermophilic enzymes exhibit high thermal stability and marked catalytic activity at high temperatures, this thermal stability is due to the robustness of the protein scaffold of thermozymes, which is thought to be less flexible than mesophilic or psychrophilic enzymes. Enzymes adapted to lower temperatures exhibit conformational flexibility, especially in the region involved in catalysis, facilitating substrate and cofactor access to binding sites and achieving high catalytic turnover at low temperatures. However, enthalpy-entropy compensation results in stronger temperature-dependent catalytic activity, leading to lower activity of thermophilic enzymes compared to mesophilic homologues at low temperatures. (DOI: 10.1016/S1095-6433(00)00359-7, DOI: 10.1146/annurev.biochem.75.103004.142723).

The industrial applications of thermozymes could be varied in processes that required high temperatures, but an interesting development related to the question of the reviewer, could be the adaptations of thermozymes to increase catalytic efficiency at lower temperatures while maintaining thermal stability. These improved enzymes could have an application to a broad set of currently set processes, where less of the more thermally stable enzymes could be required.

How to further obtain and evaluate the potential application of hydrolytic enzymes derived from isolated thermophilic bacteria?

To comprehensively obtain and evaluate the potential applications of hydrolytic enzymes derived from isolated thermophilic bacteria, it is crucial to carry out a series of methodologies. Initially, there is an emphasis on the need for a process involving the isolation and cultivation of these bacteria under conducive conditions to obtain suitable biomass, a fundamental step for the subsequent production and purification of the enzymes of interest, a central focus of this study. Subsequently, various purification techniques can be employed, such as sulfate precipitation followed by dialysis or concentration using Amicon tubes (Millipore, Merck). Essential steps include determining molecular size through SDS-PAGE gels and assessing activity in non-denaturing gels under native conditions. Once purified, a detailed characterization of their biochemical and kinetic properties is necessary, addressing aspects such as optimal pH, optimal temperature, stability, substrate specificity, and activity under different conditions. These data enable exploration of industrial applications in various fields, seeking practical benefits and leveraging the unique properties of these thermophilic enzymes. This comprehensive approach, from isolation to characterization and application exploration, contributes to a thorough and efficient understanding of the hydrolytic enzymes in question.

These methodological steps, along with the genomic analysis of the isolated bacteria, will be part of the next stages of our research. With the  genomic information in hand, data mining could be used to identify specific enzymes and produce them as recombinant proteins using mesophilic or thermophilic expression systems. We are currently working on this approach.

  1. What is the basis for determining the optimal culture temperature and temperature growth range of these isolated thermophilic bacteria in Table 3?

The study of the temperature growth range of bacteria is essential for various reasons. In biotechnological and industrial applications, understanding the optimal temperature for bacterial growth is crucial for optimizing biological processes and maximizing production efficiency. Microbiology benefits by providing valuable information about the adaptability of microorganisms to specific environments. In the production of biomass and biotechnological products, this knowledge allows for the optimization of microbial cultures, also contributing to the understanding of microbial ecology in natural environments.

In relation to our research, determining the temperature growth range of isolated bacteria also allows for their classification as strict thermophiles (not growing below 40°C). Furthermore, identifying the optimal growth temperature of the isolated bacteria enables understanding the temperatures at which maximum biomass can be achieved. In the case of certain enzymes, this correlates with the amount of enzymes produced, facilitating the production and characterization of native thermophilic enzymes.

Reviewing our data, the effect of temperature on optimal bacterial growth can be observed. For example, the SA strain displays an optimal proliferation temperature at 45ºC, at 65ºC it only reaches 50% of growth in the same growing period. Similarly, the CJ1 isolate, its optimal proliferation temperature is 65ºC; in contrast, at 55ºC, it reaches only 30% of growth after 18 hours of cultivation. Finally, the M3 strain, a maximum proliferation is observed after 22 hours of cultivation at 50ºC, compared to its maximal growth temperature at 75ºC, where it attains nearly 50% of growth during the same period of cultivation (see graphs below).

On the other hand, determining the optimal strain growth temperature represents the point at which bacteria exhibit maximum metabolic efficiency, facilitating biological processes such as DNA replication and protein synthesis. In biotechnological and industrial applications, knowing the optimal temperature is crucial for maximizing the bacterial growth rate and, consequently, biomass production. In industrial settings, maintaining conditions within the optimal range is fundamental to ensure optimal performance and consistent quality in the production of biotechnological products or in fermentative processes. The determination of the optimal temperature also plays a crucial role in the design of experiments to better understand the physiology and behavior of bacteria in different contexts.

  1. Does the selection of different buffer solutions have an impact on the optimal pH determination of isolated thermophilic bacteria?

The selection of different pH buffer solutions can impact the determination of the optimal growth pH for thermophilic bacteria. Buffer solutions work to maintain a constant pH within a specific range, and the choice of one buffer solution over another results in a change in the pH and buffer composition that changes the environmental conditions in which bacteria thrive. Different buffer solutions may have varying capacities to maintain a constant pH, influencing the ability of thermophilic bacteria to grow and function optimally at specific levels of acidity or alkalinity, but growth could be affected by the nature of the buffer composition. Our results, the growing plots at different pH showed that different strains are mainly affected by the pH and not the buffer composition.

For instance, in the case of strain CJ1, it reaches a maximum growth at pH 7, but at pH 6 and 8, its growth is nearly 90 and 80% respectively. The composition of the buffer solutions is different for pH examined (pH 6-7 HEPES, pH 8 Tris-HCl, see material and methods).  This data suggests that in this strain the proliferation is mainly affected by the pH rather than the buffer composition.

Reviewer 3 Report

Comments and Suggestions for Authors

The paper reports 15 new strains isolated from hot spring of northern Chile. This report represents a great deal of careful work to collect, isolate, and characterize these newly recovered microorganisms. The work is excellent and the paper is well-written. I fully support the publication of this work. The result are not really novel or surprising, but the work is still important and well executed.

I have one significant request / suggestion for revision. The map in figure 1 is at such a wide scale as to make it mostly useless scientifically. The authors should make a wider figure (the width of the page) that has inset local maps of the specific sampling locations so that there is an inset map of the locations for TJ, GET, QC, and then one that shows LTB/TC. If possible, there should also be lat and long coordinates of the sampling locations so that people who work with these 15 strains can be certain of where they were from.

Author Response

Reviewer 3

 I have one significant request / suggestion for revision. The map in figure 1 is at such a wide scale as to make it mostly useless scientifically. The authors should make a wider figure (the width of the page) that has inset local maps of the specific sampling locations so that there is an inset map of the locations for TJ, GET, QC, and then one that shows LTB/TC. If possible, there should also be lat and long coordinates of the sampling locations so that people who work with these 15 strains can be certain of where they were from.

We sincerely value the insightful suggestion provided by the reviewer. In response, we have meticulously crafted a new Figure 1, confident that its incorporation will substantially elevate the overall clarity of our work.

Round 2

Reviewer 1 Report

Comments and Suggestions for Authors

The authors corrected some points (construction of 16S rRNA gene-based ML phylogenetic tree, visualization of cell morphology). However other key points required to improve the paper (genome sequencing, extended characterization of isolated strains, using of advance methods for hydrolytic activities measurement) were not perfomed.

Author Response

Dear Reviewer 1,

We sincerely appreciate your comments and additional suggestions on our manuscript entitled "Isolation of thermophilic bacteria from extreme environments in Northern Chile." We greatly value the time and dedication you have invested in evaluating our work.

Regarding your suggestion to sequence the 15 genomes of the isolated bacteria, we fully understand the importance of this approach for a more comprehensive understanding of phylogenetic classification, genetic diversity, and the adaptation of microorganisms to extreme conditions. However, we would like to highlight some specific considerations that lead us not to include genomic sequencing in this particular publication.

Firstly, we want to emphasize that the main objective of our study is to inform about the process of isolating thermophilic microorganisms in the extreme environments of northern Chile. This effort has resulted in a valuable collection of cultivated strains that represent a significant resource for future research. Our primary focus in this work is to provide a detailed description of these isolation and cultivation procedures, as well as to highlight the novelty of the findings by reporting the first isolations of thermophilic bacterial cultures in some of these extreme environments in the Southern hemisphere.

Additionally, we want to note that we are committed to genomic sequencing of these strains for future studies. In fact, we have already begun sequencing two of the fifteen genomes identified in this study. However, due to resource and time limitations, we have chosen not to include this data in this particular publication. We believe that these sequencing and genomic analysis efforts deserve a more detailed attention and a comprehensive presentation in a future work specifically dedicated to this aspect of our research.

Regarding the extended characterization of isolated strains and more sophisticated methods to further description of the exhibited hydrolytic of the isolates, we would like to communicate that we are currently pursuing that, especially on those strains that have a biotechnological potential. At the moment, we are characterizing the strains Parageobacillus thermoglucosidasius C3 shows an interesting ethanol resistance and could be of utility in bioethanol generation and the Anoxybacillus flavithermus SA strain, that we are fully characterizing its amylase activity. We hope to make available this finding with the scientific community shortly.

Therefore, while we acknowledge the suggestions in the broad context of our study, we believe that its inclusion at this stage is not aligned with the objectives and scope of our current manuscript. We hope this explanation clarifies our position and justifies our decision not to follow this suggestion on this occasion.

Once again, we sincerely appreciate your comments and your contribution to improving the quality of our work. We are open to any other suggestions or questions you may have.

Sincerely,

Pedro Zamorano PhD
